# Identification and Fine-Mapping of Clubroot (*Plasmodiophora brassicae*) Resistant QTL in *Brassica rapa*

Hui Zhang [†], Xiaochao Ma [†], Xitong Liu, Shifan Zhang, Fei Li, Guoliang Li [ID], Rifei Sun *[ID] and Shujiang Zhang *

Institute of Vegetables and Flowers, Chinese Academy of Agricultural Sciences, Zhongguancun, South Street No. 12, Haidian District, Beijing 100081, China; zhanghui05@caas.cn (H.Z.); maxccaas@163.com (X.M.); liuxitong102728@163.com (X.L.); zhangshifan@caas.cn (S.Z.); lifei@caas.cn (F.L.); liguoliang@caas.cn (G.L.)
* Correspondence: sunrifei@caas.cn (R.S.); zhangshujiang@caas.cn (S.Z.)
† These authors contributed equally to this work.

**Abstract:** European fodder turnips (*Brassica rapa* ssp. *rapifera*) were identified as sources of clubroot resistance (CR) and have been widely used in *Brassica* resistance breeding. An $F_2$ population derived from a cross between a resistant turnip and a susceptible Chinese cabbage was used to determine the inheritance and locating the resistance Quantitative Trait Loci (QTLs). The parents showed to be very resistant/susceptible to the field isolates (pathotype 4) of clubroot from Henan in China. After inoculation, 27 very resistant or susceptible individuals were selected to construct bulks, respectively. Next-generation-sequencing-based Bulk Segregant Analysis Sequencing (BSA-Seq) was used and located resistance QTL on chromosome A03 (3.3–7.5 Mb) and A08 (0.01–6.5 Mb), named *Bcr1* and *Bcr2*, respectively. Furthermore, an $F_3$ population including 180 families derived from $F_2$ individuals was phenotyped and used to verify and narrow candidate regions. Ten and seven Kompetitive Allele-Specific PCR (KASP) markers narrowed the target regions to 4.3–4.78 Mb (A03) and 0.02–0.79 Mb (A08), respectively. The phenotypic variation explained (PVE) of the two QTLs were 33.3% and 13.3% respectively. The two candidate regions contained 99 and 109 genes. In the A03 candidate region, there were three candidate R genes, namely *Bra006630*, *Bra006631* and *Bra006632*. In the A08 candidate region, there were two candidate R genes, namely *Bra030815* and *Bra030846*.

**Keywords:** *Plasmodiophora brassicae*; *Brassica rapa*; QTL; resistant genes; BSA-Seq; fine-mapping

## 1. Introduction

Clubroot caused by *Plasmodiophora brassicae* Woronin has become one of most serious diseases of cruciferous crops worldwide [1]. The pathogen causes the formation of galls on the roots of susceptible plants and leads to stunted growth, wilting and premature chlorosis of the aboveground plant organs, leading to yield and quality losses. The life cycle of *P. brassicae* is thought to be a two-phase process: a primary phase occurring in the root hairs and a secondary phase occurring in the stele and cortex of the hypocotyl and roots [2,3]. In China, clubroot was identified as an important quarantine target in the first National Plant Quarantine Conference in 1953. It is distributed across most regions of the country at present [4]. *P. brassicae* is known to consist of numerous races. The differential system of Williams [5] is most commonly used to determine the races into 16, such as pathotypes 1 to 16.

BSA or QTL based on next-generation sequencing was widely used in gene mapping. Wen et al. [6] mapped heat-tolerance QTL in the tomato. Zhang et al. [7] found two QTLs in cucumber by using BSA-Seq. In Brassicae, Wang et al. [8] reported the *Brassica rapa* genome sequence. After that, a number of clubroot resistance genes/loci have been reported on seven chromosomes, A01, A02, A03, A05, A06, A07 and A08. Clubroot resistance (CR) QTL *Crr2* resistant to pathotype 2 (Pb2) was located in chromosome A01 [9]. CR QTL *CRc* resistant to Pb2 was located in A02 [10]; CR QTL *Rcr8* resistant to pathotypes 5× was located in A02 [11]. The CR loci found in chromosome A03 were totally located in three regions

of the physical map. Region I was 1.9–6.6 Mb, and CR QTL *PbBa3.1* resistant to Pb2 was mapped in this region [12]. Region II was 13.5–16.4 Mb, with CR gene *CRd* resistant to Pb4, CR QTL *Crr3* and *CRk* (QTL) resistant to Pb2 located in this region [10,13,14]. Region III was 23.59–27.23 Mb, with CR gene *CRa* resistant to Pb2; CR gene *CRb* resistant to Pb2, Pb3, Pb4 and Pb8; CR gene Rcr4 resistant to Pb2, Pb3, Pb5, Pb6 and Pb8; and Rcr5 resistant to Pb3 located in this region [11,15–17]. CR gene *CrrA5* was mapped on chromosome A05 [18]. CR QTL *Crr4* was mapped on A06 [19]. CR QTL *qBrCR38-1* resistant to pathotype7 was mapped on chromosome A07 [20]. The CR loci on chromosome A08 were totally located in two regions of physical map. Region I was 11.3–12.6 Mb; CR QTL *CRs* resistant to Pb4 [21], CR gene *Rcr3* resistant to Pb3 [22], CR gene *Rcr9* resistant to Pb5x [11] and *Crr1* resistant to Pb2 [23] were mapped in this region. Region II was 20.2–21.7 Mb; CR QTL *qBrCR38-2* resistant to Pb7 was located in this region [20]. Among the mapped genes/loci mentioned above, *CRa* and *CRb* on A03 and *Crr1* on A08 were cloned, and they are all R genes. Makers linked to these genes were designed and used in breeding.

In this study, an $F_2$ population derived from a cross between a resistant turnip and a susceptible Chinese cabbage was used to identify resistance QTL. Next-generation sequencing-based BSA-Seq was used to locate resistance QTL. SNP-index and $\Delta$(SNP-index) graphs were identified base on bioinformatics information of the *Brassica rapa* genome. Two regions on chromosome A03 and A08 showed significant differences, respectively.

## 2. Materials and Methods

### 2.1. Plant Materials and Pathogen Isolates

An $F_2$ population (n = 206) of *B. rapa* was developed by crossing '877' and '255'. The parental genotype '877' is a European fodder turnip (*Brassica rapa* ssp. *Rapifera*) inbred line, and its highly resistance to clubroot disease originated from ECD04 (*B. rapa rapifera*) [24]. The parental genotype '255', which is highly susceptible to clubroot, is a Chinese cabbage inbred line. One hundred and eighty $F_3$ families from $F_2$ individuals were used to confirm candidate areas. *P. brassicae* field isolates were collected from Henan province (Pbh) in China, and stored at $-20\,^{\circ}$C until required.

### 2.2. Inoculation

Seeds of the parents, $F_1$, $F_2$ and $F_3$ were surface-disinfected in 1% sodium hypochlorite for one min, washed with distilled water and germinated in 12 cm–diameter Petri dishes on moistened filter paper for five days at room temperature. Preparation of *P. brassicae* resting spores was modified from Feng et al. [25]. Briefly, the frozen galls were thawed at room temperature and ground in a blender with distilled water. The resulting homogenates were passed through eight layers of cheesecloth, and the filtrate was centrifuged at $2000 \times g$ for five min. The pellet containing the *P. brassicae* resting spores was suspended in distilled water. The concentration of resting spores was measured with a haemocytometer and adjusted to $1 \times 10^8$ resting spores per mL with sterile distilled water.

### 2.3. Evaluation of Clubroot Resistance

Two parents (30 plants), $F_1$ (30 plants), $F_2$ (206 plants) and $F_3$ (180 families) populations were phenotyped against Pbh. For testing of $F_{2:3}$ families, the trial was conducted in randomized blocks design with 3 repetitions and 10 plants in each repetition. The seedlings were inoculated with the resting spores by immersing the roots into the inoculum suspension for one minute. The plants were then immediately transplanted into autoclaved potting medium in 6 cm $\times$ 6 cm $\times$ 6 cm plastic pots, at a density of one seedling per pot. The pots were kept in water-filled trays for 14 d (to ensure sufficient moisture for infection by *P. brassicae*), after which they were transferred to a bench, where they were watered from above. The plants were evaluated for clubroot disease severity six weeks after inoculation [26]. The roots were rated on a 0–9 scale (0 = no infection; 9 = heavily infested) (Figure 1) [27]. A disease index (DI) was calculated according to the formula $DI = [(0n_0 + 1n_1 + \ldots + 9n_9) \times 100]/(9 \times N_T)$.

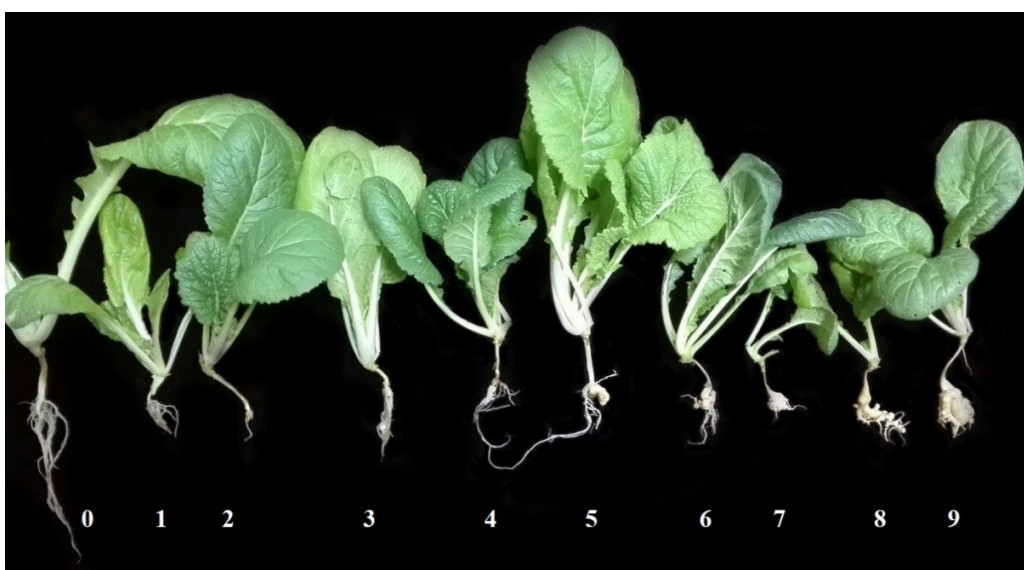

**Figure 1.** Disease rating scale ranges of clubroot symptoms on plant roots in $F_2$ population. Scale is from 0 to 9, where 0 = no infection (no symptoms) and 9 = heavily infected (severe galling).

## 2.4. DNA Sequencing

Genomic DNA was extracted from fresh leaves, using the CTAB (cetyltrimethyl-ammonium-bromide) [28] method. Each one of the twenty-seven very resistant (R)/susceptible (S) individuals was selected to construct R/S bulks, respectively. A total amount of 1.5 μg DNA per sample was used as input material for DNA sample preparations. The DNA sample was fragmented by sonication to a size of 350 bp; then DNA fragments were end-polished, A-tailed and ligated with the full-length adapter for Illumina sequencing with further PCR amplification. At last, PCR products were purified (AMPure XP system), and libraries were analyzed for size distribution by Agilent2100 Bioanalyzer and quantified by using real-time PCR. Parents and bulks genomic DNA were sequenced by Illumina HiSeq4000 platform, and 150 bp paired-end reads were generated with insert size around 350 bp.

## 2.5. Data Analysis

To make sure that reads were reliable and without artificial bias (low-quality paired reads, which mainly resulted from base-calling duplicates and adapter contamination) in the following analyses, raw data (raw reads) of fast format were firstly processed through a series of quality control (QC) procedures in-house C scripts. BWA (Burrows–Wheeler Aligner) [29] was used to align the clean reads of each sample against the reference genome. Alignment files were converted to BAM files by using SAMtools software [30]. In addition, potential PCR duplications were removed by using SAMtools command "rmdup". If multiple read pairs have identical external coordinates, only retain the pair with the highest mapping quality. Variants calling was performed for all samples by using the Unified Genotyper function in GATK software. Single-Nucleotide Polymorphisms (SNPs) were used as Variant Filtration parameter in GATK [31] (settings: –filterExpression "QD < 4.0 ‖ FS > 60.0 ‖ MQ < 40.0", -G_filter "GQ < 20", –clusterWindowSize 4). Insertion–deletion (InDel) was filtered by using the Variant Filtration parameter (settings: –filter Expression "QD < 4.0 ‖ FS > 200.0 ‖Read PosRankSum < −20.0 ‖ Inbreeding Coeff < −0.8"). ANNOVAR [32], an efficient software tool, was used to annotate SNP based on the GFF3 files for the reference genome.

## 2.6. SNP/InDel Index

The homozygous SNPs/InDels between two parents were extracted from the vcf files for SNP/InDel. The reads depth information for homozygous SNPs/InDels above in the

offspring pools was gained to calculate the SNP/InDel index [33]. We used the genotype of one parent as the reference and the statistic reads number for this parent's genotype or the others in offspring pool. Then we calculated the ratio of the number of different reads in total number, which is the SNP/InDel index of the base sites. We filtered out those points for which the SNP/InDel index in both pools are less than 0.3. Sliding-window methods was used to present the SNP/InDel index of whole genome. The average of all SNP/InDel index in each window was as the SNP/InDel index for this window. Usually, we used a window size of 0.5 Mb and step size of 5 Kb as default settings. The difference of the SNP/InDel index of two pools was calculated as the delta SNP/InDel index.

### 2.7. Development of KASP Markers and QTL Mapping

To validate the BSA-Seq results, KASP (Kompetitive Allele Specific PCR) markers were designed, using Primer 6.0 software (http://www.premierbiosoft.com/, accessed on 18 October 2020), by the Laboratory of the Government Chemist base on the *Brassica rapa* genome v1.5 (http://brassicadb.cn/#/, accessed on 23 October 2020). All the primers were synthesized by Sangon Biological and Engineering Co. (Shanghai, China). The genotype consistent with parent '877' is marked as 'A', the genotype consistent with parent '255' is marked as 'B', the heterozygous genotype is marked as 'H' and the missing genotype is marked as '-'. Linkage analysis was performed by JoinMap v4.0 [34] to the Kosambi function, and the maximum-likelihood method was used to calculate the genetic distance. Map QTL v4.0 was used to detect the QTL, using the Interval Mapping (IM) and Multiple QTL Mapping (MQM) pattern under a threshold of LOD = 2.0.

## 3. Results

### 3.1. Test of Clubroot Resistance

The resistance of parents and $F_1$ were tested against Pbh. The resistant parent '877' was highly resistant (DI = 0) to the isolate, and the susceptible parent '255' was highly susceptible to Pbh (DI > 95); $F_1$ plants showed an intermediate DI value between two parental lines (DI = 38.1). Two hundred and six $F_2$ individuals were tested by Pbh, and 199 individuals were evaluated. In the $F_2$ population, more than half of the individuals were scaled as '0' or '9' grades (70 individuals' grades were '0', and 56 individuals' grades were '9'), and 73 individuals were scaled as '1' to '8' (Table 1). Thus, it was suggested that the clubroot resistance was controlled by major QTL with complementary effect. The $F_3$ population, including 180 families derived from another $F_2$ population, was inoculated by Pbh. The DI of each $F_{2:3}$ family was calculated. One hundred and sixty-four families received the DI values (Supplementary Materials Table S1); the resistance of $F_2$ individuals was obtained from their $F_{2:3}$ family (Figure 2).

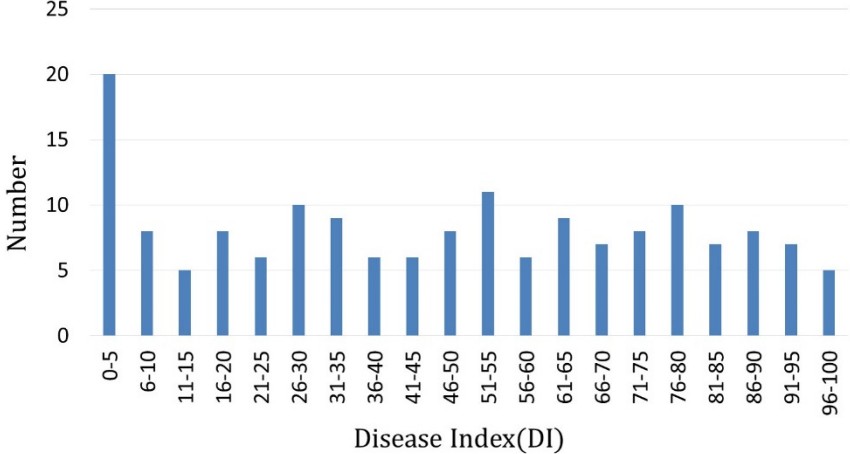

**Figure 2.** Distribution of $F_{2:3}$ families' disease levels (disease index) after testing with Pbh of *P. brassicae*.

**Table 1.** Distribution of F₂ disease levels (disease index) after testing with Pbh of *P. brassicae*.

| | Scale for Rating Clubroot Symptoms | | | | | | | | | |
|---|---|---|---|---|---|---|---|---|---|---|
| | 0 | 1 | 2 | 3 | 4 | 5 | 6 | 7 | 8 | 9 |
| Numbers of F₂ Individuals | 70 | 22 | 7 | 10 | 5 | 5 | 7 | 12 | 5 | 56 |

*3.2. Sequencing Data Analysis*

Next-generation-sequencing-based BSA-Seq was used to locate resistance genes. Each of the twenty-seven very resistant (scale = 0)/susceptible (scale = 9) F₂ individuals was selected to construct R/S bulks, respectively. Sequencing data were generated with Illumina HiSeq 4000, with an average insert size of around 350 bp. A total of 6,402,752,700, 6,158,942,400, 14,984,101,500 and 15,440,598,600 raw data were obtained from parent '877', parent '255', R bulk and S bulk, respectively. Clean data were obtained after removing adapter-polluted and low-quality reads and unknown bases (N > 5%). Genome coverage ranged around 90%, and average depths were 12.17×, 18.64×, 28.67× and 30.08× for '877', '255', and R bulk and S bulk, respectively (Table 2). The genotype of '255' was used as the reference and to statistic reads number for this parent's genotype or the others in R/S bulks. After filtered, 979,164 SNP indexes were obtained. The average number of sequence variations on 10 chromosomes was 108,138, with chromosome A09 having the highest and A10 having the lowest number of variations. Chromosome A03 had a high density of variations in a specific region (Figure 3).

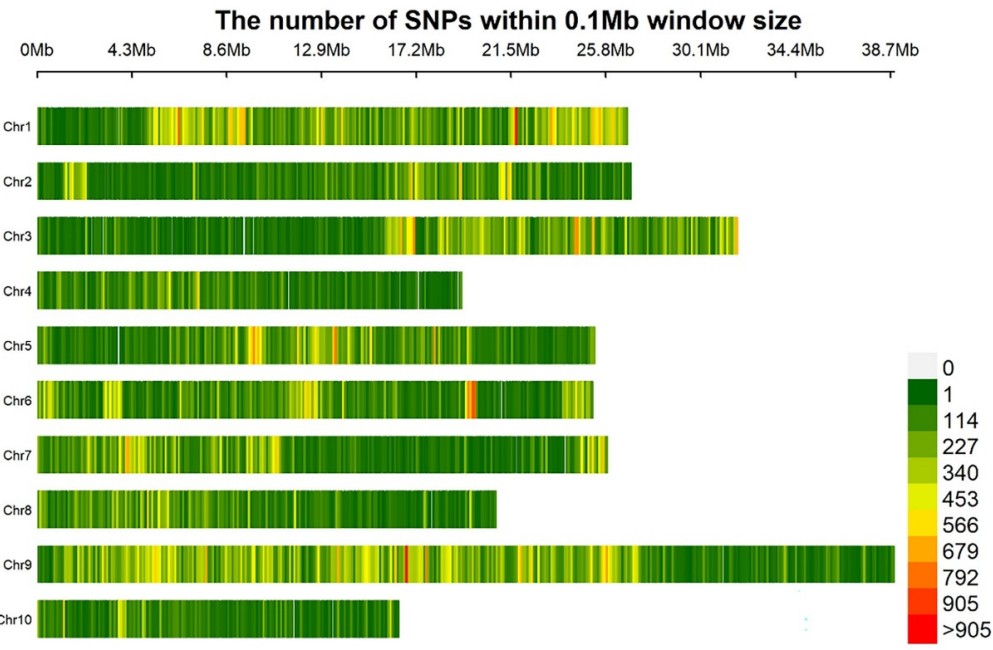

**Figure 3.** Distribution density map of SNPs on each chromosome within a sliding window of 100 kb. Note: Ordinates represent chromosomes 1 to 10, the abscissa represents the physical location of the chromosome and different colors show the density of SNPs within a sliding window of 100 kb.

**Table 2.** Depth and coverage of re-sequencing data.

| Sample | Mapped Reads | Total Reads | Mapping Rate (%) | Average Depth(×) | Coverage at Least 1× (%) | Coverage at Least 4× (%) |
|---|---|---|---|---|---|---|
| PS | 32,638,843 | 40,655,484 | 80.28 | 12.17 | 91.83 | 75.90 |
| PR | 38,402,704 | 42,580,738 | 90.19 | 18.64 | 88.66 | 83.76 |
| S-bulk | 71,462,576 | 102,510,534 | 69.71 | 28.67 | 95.50 | 92.63 |
| R-bulk | 82,274,316 | 99,471,594 | 82.71 | 30.08 | 94.89 | 88.61 |

### 3.3. Association Analysis

The sliding-window methods was used to present the SNP index between R bulk '877' and S bulk '255', respectively. The difference of the SNP index of two bulks was calculated as the ΔSNP index (Figure 4). The window size was described above. Two regions on chromosome A03 and A08 showed significant differences, respectively. One region was on chromosome 3, named *Bcr1* (*Brassica rapa* clubroot resistance 1), at 3.3–7.5 Mb, and the other region was on chromosome A08, named *Bcr2* (*Brassica rapa* clubroot resistance 2), at 0.01–6.5 Mb. The absolute values of the SNP-index of *Bcr1* and *Bcr2* were greater than the threshold and were close to 0.5 at a confidence level of 95%. Observations of SNP haplotypes among the highly resistant plants in the R pool were the same as those in the parent '877', while highly susceptible plants in the S pool contained alleles from the parent '255', thus indicating that there have been major QTLs controlling swollen roots in these regions.

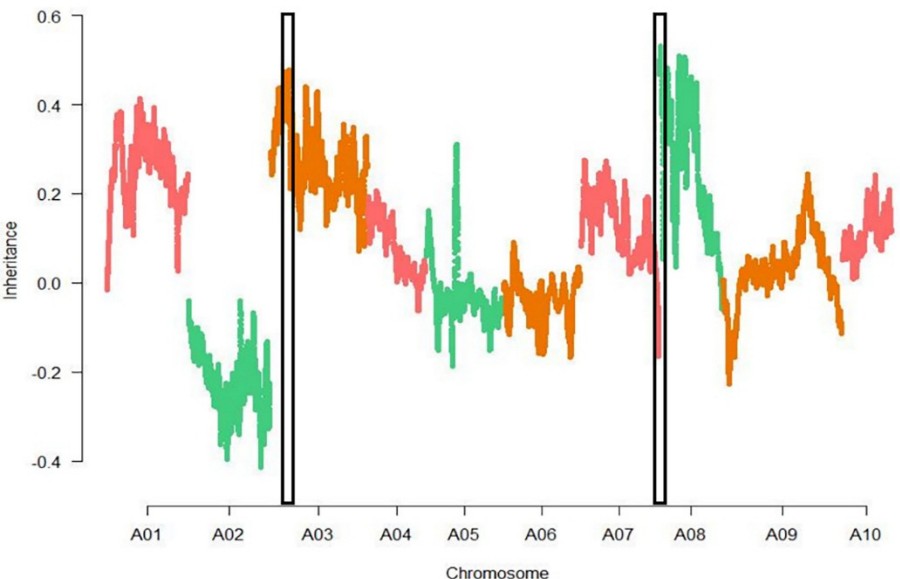

**Figure 4.** Genome−wide ΔSNP index Manhattan plots graph of two extreme pools. Two regions on chromosome A03 and A08 (black box) were identified.

### 3.4. Fine-Mapping of Two QTL

Based on the whole-genome sequencing data of the parents, SNP sites were selected every 20~200 kb at target regions for primer design. Competitive allele-specific PCR (KASP: Kompetitive Allele Specific PCR) was conducted in parents, and $F_1$ was used to detect the selected SNP markers. Seventy-eight and 39 pairs of KASP makers with better typing effects were selected from candidate regions, respectively (Supplementary Materials Table S2). At the same time, twelve individuals from R bulk and 12 individuals from S bulk were verified with selected KASP markers. Most polymorphic markers showed 'A' genotype ('877' genotype) and 'H' genotype (heterozygous genotype) in the R bulk, and most polymorphic markers showed 'B' genotype ('255' genotype) and 'H' genotype in the S bulk, thus proving that the selected markers have a certain linkage relationship with phenotype; they were used for subsequent test analysis. The *Bcr1* and *Bcr2* regions were verified and fine-mapped. One hundred and sixty-four individuals from the $F_{2:3}$ population were genotyped. JoinMap 4.0 (OOIJEN and VAN, 2006) was used for linkage analysis. Seventy-eight and 39 pairs of KASP makers were used for constructing genetic maps on A03 and A08 target regions, respectively. Interval Mapping (IM) combined with Multiple QTL Mapping (MQM) method of Map QTL (OOIJEN et al., 2009) software was used for detecting QTLs related to disease-resistance traits. Among the mapping markers, ten and seven KASP markers were used for mapping.

*Bcr1* was narrowed to 3.3–7.5 Mb and 4.3–4.78 Mb in the A03 chromosome between markers A03-1-192 and A03-1-024; the likelihood of odd (LOD) value of A03-1-115 was the highest, at 14.4. *Bcr1* explained 33.3% of the phenotypic variation of resistant to clubroot (Figure 5). There were 99 genes in the *B. rapa* genome (http://brassicadb.cn/#/, accessed on 23 October 2020), covering 480 kb on chromosome A03, which is homologous to chromosome 5 in *Arabidopsis*. Three genes (*Bra006630*, *Bra006631* and *Bra006632*) were identified in tandem array, which encodes TIR–NBS–LRR protein in the candidate region. (Table 3).

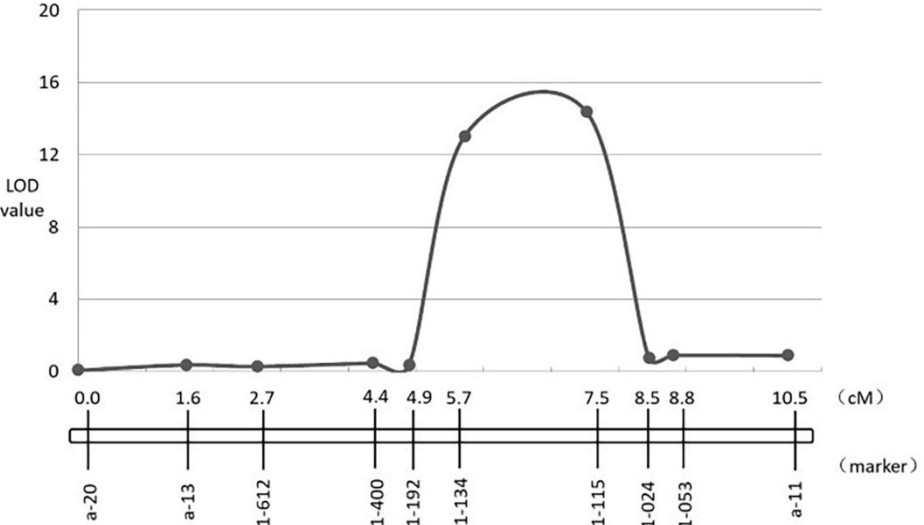

**Figure 5.** Identification and validation of CR QTL *Bcr1* on chromosome A03.

**Table 3.** Candidate R genes in candidate regions.

| Gene ID | Chromosome Position | Start Position of the Gene | End Position of the Gene | Orthologous Genes | Gene Annotations |
|---|---|---|---|---|---|
| *Bra006630* | A03 | 4,355,675 | 4,357,301 | *AT5G22670* | FBD; leucine-rich repeat 2; FBD-like; cyclin-like F-box; phosphoglycerate/bisphosphoglycerate mutase, active site |
| *Bra006631* | A03 | 4,365,925 | 4,367,585 | *AT5G22670* | FBD; leucine-rich repeat 2; FBD-like; cyclin-like F-box |
| *Bra006632* | A03 | 4,370,728 | 4,372,332 | *AT5G22730* | FBD; leucine-rich repeat 2; cyclin-like F-box |
| *Bra030815* | A08 | 57,170 | 62,428 | *AT1G56130* | Serine-threonine/tyrosine-protein kinase, catalytic domain; leucine-rich repeat; malectin domain |
| *Bra030846* | A08 | 219,222 | 222,719 | *AT1G55610* | Protein kinase domain; leucine-rich repeat; leucine-rich repeat-containing N-terminal, plant-type |

*Bcr2* was narrowed from the range of 0.01–6.5 Mb to the range of 0.02–0.79 Mb in A08 chromosome between markers A08-1-06 and A08-1-705; the likelihood of odd (LOD) value of A08-1-348 was the highest, at 4.92. *Bcr2* explained 13.3% of the phenotypic variation of resistant to clubroot (Figure 6). There were 109 genes in the *B. rapa* genome (http://brassicadb.cn/#/, accessed on 23 October 2020), covering 770 kb on chromosome A08, which is homologous to chromosome 1 in *Arabidopsis*. Two genes (*Bra030815* and

*Bra030846*) were identified which encode TIR–NBS–LRR protein the in candidate region (Table 3).

**Figure 6.** Identification and validation of CR QTL *Bcr2* on chromosome A08.

## 4. Discussion

The European fodder turnip was thought to carry broad-spectrum resistance [35,36]. In this study, the European fodder turnip '877' originating from ECD04 was used as the resistant parent. The turnip '877' was resistant to all isolates that were collected from Yunnan (Pb4, pathotypes 4), Henan (Pb4), Beijing (Pb4 and Pb5), Hubei (Pb2 and Pb4) and Sichuan (Pb4) provinces in China and single-spore isolates Pb3, Pb5 and Pb6 in previous studies (test by Williams's differential set) [5], which indicated that '877' had resistant genes for different pathotypes and isolates. The isolate from Henan (Pbh) used in this study had the highest virulence in all P4 isolates. $F_2$ and $F_{2:3}$ families were used for mapping and fine-mapping, and two QTL were obtained. In the $F_2$ mapping population, more than half of the individuals were scaled as '0' or '9', so the resistant loci were also analyzed by using the quality trait method. Individuals graded '0' to '3' (Table 1) were regarded as 'resistant', because the galls were small and all on lateral roots. Individuals graded '4' to '9' (Table 1) were regarded as 'susceptible', because galls were found on the main root. One hundred ninety-nine individuals, including 109 resistant plants (scale 0–3) and 90 susceptible plants (scale 4–9), were consistent with the ratio of 9:7 segregation ($\chi^2 = 0.12 < \chi^2_{0.05} = 3.84$), thus indicating that resistance was controlled by two genes. In the $F_{2:3}$ population, $F_2$ individuals were regarded as 'resistant' when their $F_3$ family's DI < 50, the rest of the $F_2$ individuals were regarded as 'susceptible' to Pbh (Figure 2). Eighty-six $F_{2:3}$ families were resistant to Pbh, and 78 $F_{2:3}$ families were susceptible to Pbh. The segregating ratio of resistance and susceptibility was also consistent to 9:7 ($\chi^2 = 0.05 < \chi^2_{0.05} = 3.84$). The segregating ratio of these two populations was 9:7, which indicated that there may be two major genes controlling the resistance. This result was similar with the result in QTL mapping.

In this study, the genomes of '877' and '255' were also re-sequenced, and 979,164 and 102,222 SNPs and InDels with polymorphisms between the parents were obtained. According to the analysis results of QTL-Seq, we developed a large number of SNP markers in the candidate interval to detect the accuracy of *Bcr1* and *Bcr2* sites and to further fine-map them. Among them, the SNP marker can detect the difference of a single nucleotide, and its distribution on the genome is more extensive; and the SNP marker can be typed by KASP technology, which is one of the SNP typing platforms that has now developed into a global

SNP score. This type of benchmark technology has the advantages of high throughput, low cost, flexibility and speed.

According to the BSA-Seq, the resistant QTLs were found in this study on chromosome A03 and A08. There were eleven genes/loci mapped on chromosome A03 and five on A08 [37]. The CR loci in A03 were totally located in three regions. Region I was 1.9–6.6 Mb, Region II was 13.5–16.4 Mb and Region III was 23.59–27.23 Mb. The clubroot resistance (CR) gene *CRa* resistant to Pb2 originated from the European fodder turnip ECD02 [38]; the CR gene *CRb* resistant to Pb2, Pb3, Pb4 and Pb8 originated from the European fodder turnip ECD01 [39]; the CR locus *CRk* resistant to Pb2 from Region II originated from the European fodder turnip Debra; the CR locus *Crr3* resistant to Pb2 originated from European fodder turnip Milan White; and CR QTL *PbBa3.1* resistant to Pb2 from Region I originated from European fodder turnip ECD04. *Bcr1* in this study also originated from ECD04 and is located in Region I, but it is resistant to Pb4, in contrast to *PbBa3.1* resistant to Pb2. The CR loci in A08 were totally located in two regions. Region I was 11.3–12.6 Mb, and Region II was 20.2–21.7 Mb. *Crr1* resistant to Pb2 from Region I originated from 'siloga' [19]; *Rcr3* resistant to Pb3 [22] and *Rcr9* resistant to Pb5x [11] originated from 'Waaslander'; *qBrCR38-2* resistant to Pb7 [20] and mapped in Region II originated from 'Maikno' (pakchoi). *Bcr2* in this study was mapped on 0.02–0.79 Mb in chromosome A08, which is a new locus in A08.

According to the results of fine-mapping, there were three and two R genes (encoding Nucleotide-Binding domain and Leucine-Rich Repeats, NBS–LRR, proteins) in candidate regions. The previously cloned resistant genes *CRa*, *CRb* and *Crr1* all encode TIR–NBS–LRR protein [17,40]. *Bra006630*, *Bra006631* and *Bra006632* in A03 were arranged in a tandem array. These five R gene in two regions will be the prior target for the next study for confirming candidate genes.

## 5. Conclusions

In this study, an $F_2$ population derived from a cross between a resistant turnip and a susceptible Chinese cabbage was used to determine the inheritance and location of the resistance QTL. The parents were determined to be very resistant/susceptible to the field isolates (pathotype 4) of clubroot from Henan in China. BSA-Seq was used and located resistance QTLs on chromosome A03 (3.3 Mb–7.5 Mb) and A08 (0.01 Mb–6.5 Mb), named *Bcr1* and *Bcr2*, respectively. Furthermore, an $F_3$ population derived from $F_2$ individuals was phenotyped and used to verify and narrow candidate regions. Ten and seven KASP markers narrowed the target regions to 4.3–4.78 Mb (A03) and 0.02–0.79 Mb (A08), respectively. The phenotypic variation explained (PVE) of the two QTLs were 33.3% and 13.3%, respectively. The two QTLs contained 99 and 109 genes, covering 480 kb on chromosome A03 and 770 kb on A08. Based on the genes annotation, there were three and two candidate R genes in the candidate regions. These R genes were not reported to be associated with clubroot resistance.

**Supplementary Materials:** The following supporting information can be downloaded at: https://www.mdpi.com/article/10.3390/horticulturae8010066/s1. Table S1: The results of clubroot resistance identification of 164 $F_{2:3}$ families. Table S2: The information of KASP primer sequences in candidate regions.

**Author Contributions:** Conceptualization, H.Z.; data curation, X.M.; formal analysis, H.Z. and X.M.; funding acquisition, S.Z. (Shujiang Zhang); investigation, X.L.; methodology, S.Z. (Shifan Zhang); project administration, R.S. and S.Z. (Shujiang Zhang); resources, H.Z.; software, G.L.; supervision, R.S.; validation, F.L.; visualization, X.M.; writing—original draft, H.Z.; writing—review and editing, R.S. and S.Z. (Shujiang Zhang). All authors have read and agreed to the published version of the manuscript.

**Funding:** This research was supported by China Agricultural Research System (CARS-23-A-14) and the Agricultural Science and Technology Innovation Program of the Chinese Academy of Agricultural Sciences (CAAS-ASTIP-IVFCAAS). This study was carried out at the Key Laboratory of Biology and Genetic Improvement of Horticultural Crops, Ministry of Agriculture, Beijing, China.

**Institutional Review Board Statement:** Not applicable.

**Informed Consent Statement:** Not applicable.

**Data Availability Statement:** Sequence data can be found in the NCBI databases under the following accession numbers: PR1: SRR16016164, PR1-mix: SRR16016163, PS1: SRR16016162 and PS1-mix: SRR16016161.

**Acknowledgments:** The authors are thankful to Feng Cheng, Xiaowu Wang and Jian Wu (Institute of Vegetables and Flowers, Chinese Academy of Agricultural Sciences) for data analysis, laboratory assistance and helpful comments about the study.

**Conflicts of Interest:** The authors declare no conflict of interest.

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
