# Peer review of "Identification and Fine-Mapping of Clubroot (Plasmodiophora brassicae) Resistant QTL in Brassica rapa"

_horticulturae, doi:10.3390/horticulturae8010066_

Round 1

Reviewer 1 Report

Clubroot (CR) caused by Plasmodiophora brassicae is one of the most serious diseases of Brassicaceae worldwide. In this article, the authors used BSA-seq with KASP analysis to conduct the mapping. As a result, two QTLs regions on chromosomes A03 and A08 respectively were detected, and five genes in the target chromosome region associated with the disease resistance were considered as candidates for CR. These genes need to be further studied, and the associated SNPs should be used for marker-assisted breeding of CR in B. rapa.

However, I have some suggestions/shortcomings for the authors and requested to be addressed these shortcomings prior to publication.

Line 15: QTL (Please write the full form).

Line 18: BSA–seq (Please write the full form).

Line 21: KASP (Please write the full form).

Line 42: Please rewrite this line ‘Rcr8 (QTL) resistant to Pb5X was mapped on A 02 either’.

In the Introduction please write about Next-generation sequencing-based BSA–seq methods briefly and includes references where previously used and which crops.

Line: 90-91- Please add the reference.

Figure 1: I am not sure about this picture of parents or F2. If so, please indicate that and mark like that 0-4 resistant, 5-9 susceptible line. Also, indicate scoring number direction from left to right or number 0,1,2…...

Line 96: for CTAB - please add a reference.

Line 123-24: Please write a full form of SNPs/InDels.

Line 129-130: Please add the reference.

Line 150: Please confirm how many isolates were used in this study. I understand only one from Henan province (Pbh) in China or three?

Figure 2: Please give Table instead of Figure. It will be more visible.

Figure 3: Please confirm, I think it will be 0-5, 6-10……

Line 168--: Please give the summary of BSA-seq or QC of sequencing data for each parent and /or R/S pool in the table that you discussed in section 3.2.

Figure 4: Please clearly indicate the meaning of horizontal, vertical and different colors in the figure legend.

Line 187: Brassica rapa clubroot resistance 2 (Not 1)

Line 206: Please omit one genotype.

Line 266-269: Please confirm which scaling is correct.

‘Individuals graded ‘0’ to ‘4’ were regarded as ‘resistant’, because the gall was small and all on lateral roots. Individuals graded ‘5’ to ‘9’ were regarded as ‘susceptible’ because galls were found on the main root. One hundred ninety-nine individuals, including 109 resistant plants (scale 0-3) and 90 susceptible plants (scale 4-9).

Line 270-274: I think you did the Genetic analysis of F2 population. Depending on the data you discussed here but the result does not show in the Result section or a supplementary section. Please add this data.

Line 282: fine- locate them?

Line 305-308: Bra006630, Bra006631, and Bra006632 in A03 were arranged in a tandem array- what about the other two genes in A08?

The sequence of Bra006632 was blast between parents- Why not the other two genes?

16 bp delection? This deletion has any effect for making resistance to CR. What does the author think about that?

References

Please follow the rules of the paper for writing references and check carefully each reference.

I think the reference section authors do not check carefully.

The year will come after Journal Name. I checked only 5 references and I found mistakes in all.

  1. Plasmodiophora brassicae – Italic.
  2. Journal.-J.
  3. Plasmodiophora brassicae. – Italic.
  4. J. Plant Sci - Can. J. Plant Pathol. And Plasmodiophora Brassicae - Italic.
  5. 107(6): 107; Theor. Appl. Genet. – journal name should be Italic.

Overall, authors did the good work. I am requesting to improve the manuscript following comments and checking the English carefully.

Author Response

Response to Reviewer 1 Comments

Point 1:  Line 15: QTL (Please write the full form).

Response 1: Full form of QTL was added as “Quantitative Trait Loci”.

Point 2: Line 18: BSA–seq (Please write the full form).

Response 2: Full form of BSA-seq was added as “Bulk Segregant Analysis sequencing”.

Point 3: Line 21: KASP (Please write the full form).

Response 3: Full form of KASP were added as “Kompetitive Allele-Specific PCR”.

Point 4: Line 42: Please rewrite this line ‘Rcr8 (QTL) resistant to Pb5X was mapped on A 02 either’.

Response 4: This line was rewritten.

Point 5: In Introduction please write about Next generation sequencing based BSA–seq methods briefly and includes references where previously used and which crops.

Response 5:  The methods using in gene mapping was add in INTRODCUTION with references.

Point 6: Line: 90-91- Please add the reference.

Response 6: Reference were added.

Point 7: Figure 1: I am no sure this picture of parents or F2. If so, please indicate that and marking like that 0-4 resistant, 5-9 susceptible line. Also indicate scoring number direction from left to right or number 0,1,2…...

Response 7: This figure were shown rating scales of clubroot in F2 population. The numbers were added in Figure 1 and F2 was added in figure legend.

Point 8: Line 96: for CTAB - please add reference.

Response 8: Reference were added.

Point 9: Line 123-24: Please write full form of SNPs/InDels.

Response 9: Full forms were added as “Single Nucleotide Polymorphisms” and “Insertion-delection”.

Point 10: Line 129-130: Please add the reference.

Response 10: Reference were added.

Point 11: Line 150: Please confirm how many isolates used in this study. I understand only one from Henan province (Pbh) in China or three?

Response 11: The isolate were revised. “three isolates” was deleted. Only Pbh was used for QTL mapping.

Point 12: Figure 2: Please give Table instead of Figure. It will be more visible.

Response 12: Figure 2 was changed to Table 1.

Point 13: Figure 3: Please confirm, I think it will be 0-5, 6-10……

Response 13: Figure 3 (Figure 2 now) was revised.

Point 14: Line 168--: Please give the summery of BSA-seq or QC of sequencing data for each parents and /or R/S pool in table that you discussed in the section 3.2.

Response 14: Table 2 was added.

Point 15: Figure 4: Please clearly indicate what the meaning of horizontal, vertical and different color in figure legend.

Response 15: The figure legend of Figure 4 (Figure 3 now) was revised.

Point 16: Line 187: Brassica rapa clubroot resistance 2 (Not 1)

Response 16: revised

Point 17: Line 206: Please omit one genotype.

Response 17: revised

Point 18: Line 266-269: Please confirm which scaling is correct.

‘Individuals graded ‘0’ to ‘4’ were regarded as ‘resistant’, because the gall were small and all on lateral roots. Individuals graded ‘5’ to ‘9’ were regarded as ‘susceptible’, because galls were found on the main root. One hundred ninety-nine individuals, including 109 resistant plants (scale 0-3) and 90 susceptible plants (scale 4-9).

Response 18: This part was revised as ‘0’ to ‘3’ were gregarded as ‘resistant’ and ‘4’ to ‘9’ were regarded as ‘susceptible’. Data was shown in Table 1.

Point 19: Line 270-274: I think you did the Genetic analysis of F2 population. Depending on data you discussed here but the result does not show in Result section or supplementary section. Please add this data.

Response 19: Brc1 and Brc2 were found and mapped in this research base on BSA-seq and QTL mapping. These two loci were treated as quantitative traits. So we just presented the QTL mapping results in RESULTS part.  Genetic analysis is an analysis method based on quality traits. The resistant loci can not be treated as quantitative traits and quality traits both in RESULTS part. So we discuss these two different methods in DISCUSSION part, because they had similar results. We revised and explain more in DISCUSSION part.

 Data were used in genetic analysis was the same as QTL mapping data which showed in Table 1, Figure 2. And the segregation ratio was shown in DISCUSSION part.

Point 20: Line 282: fine- locate them?

Response 20: Revised as “fine mapping”

Point 21: Line 305-308: Bra006630, Bra006631 and Bra006632 in A03 were arranged in a tandem array- what about the other two genes in A08?

Response 21: Two candidate genes in A08 were not arranged in tandem array.

Point 22: The sequence of Bra006632 were blast between parents- Why not other two genes?

16 bp delection? This deletion has any effect for making resistant to CR. What the author thinks about that?

Response 22: This sentence was deleted. We cloned Bra006632, so the sequences were blast between parents. But these work was not finished yet (so we delete this sentence). And we supposed the resistance may controlled by these three R genes not one of them in A 03. Cloning and qPCR detection of these five genes in A03 and A08, and the genes functions will be the next important work based on these results.

Point 23: References

Please follow the rules of the paper for writing references and check carefully each reference.

I think the reference section authors does not check carefully.

Year will be come after Journal Name. I checked only 5 references and I found mistake in all.

  1. Plasmodiophora brassicae – Italic.
  2. - J.
  3. Plasmodiophora brassicae. – Italic.
  4. J. Plant Sci - Can. J. Plant Pathol. And Plasmodiophora Brassicae - Italic.
  5. 107(6): 107; Theor. Appl. Genet. – journal name should be Italic.

Response 23: The formats of references were checked and revised.

Reviewer 2 Report

Review ID 1516912

 Identification and fine mapping of clubroot (Plasmodiophora 2 brassicae) resistant QTL in Brassica rapa

               This is quite well organized manuscript. I found this “ms” interesting and innovative. However, a few questions must be explained more precisely.

Critical review:

  1. Introduction is unacceptable. The presented text should reach every reader. This is not cryptography, just a scientific article, but also covering every reader not necessarily related to science.
  2. Methodology is well presented.
  3. Figures 2 and 3 do not represent the scientific path.
  4. I don't understand the presentation in Table 1. What does it bring? You can fill the next two pages with the codes too. Is it to highlight the article? This should be easy to read by different readers. Otherwise why is it for?
  5. Less codes and more references to other publications please.
  6. No conclusion presented.

Author Response

Response to Reviewer 2 Comments

Point 1: Introduction is unacceptable. The presented text should reach every reader. This is not cryptography, just a scientific article, but also covering every reader not necessarily related to science.

Response 1: INTRODUCTION was improved.

Point 2: Figures 2 and 3 do not represent the scientific path

Response 2: Figure 2 was changed to Table 1. Figure 3 (Figure 2 now) was adjusted.

Point 3: I don't understand the presentation in Table 1. What does it bring? You can fill the next two pages with the codes too. Is it to highlight the article? This should be easy to read by different readers. Otherwise why is it for?

Response 3: Table 1 showed the primers sequences of KASP markers using in QTL mapping. We pretend to provide our primers information directly to other researchers for potential needed. Complete information for all the primers was in Table S2. If table 1 not necessary, then we deleted it.  

Point 4: Less codes and more references to other publications please

Response 4: The codes like qBrCR38-2 in introduction and discussion are genes or loci name in other publications. In these two part we did more explanation and added more references.

Point 5: No conclusion presented.

Response 5:  CONCLUSION was added

Round 2

Reviewer 1 Report

Thank you for addressing the information. But I think before publishing it’s should be improved in the reference section.

Line 83: P. brassicae should be italic.

Please check very carefully the reference part again.

Like reference number 6,7,8 and so on.